# Investigating stratospheric changes between 2009 and 2018 with trace gas data from aircraft, AirCores, and a global model focusing on CFC-11

Johannes C. Laube[1,2]*, Emma C. Leedham Elvidge[2,3] Karina E. Adcock[2], Bianca Baier[4,5], Carl A.M. Brenninkmeijer[6], Huilin Chen[7], Elise S. Droste[2], Jens-Uwe Grooß[1], Pauli Heikkinen[8], Andrew J. Hind[2], Rigel Kivi[8], Alexander Lojko[2,9], Stephen A. Montzka[5], David E. Oram[2], Steve Randall[10], Thomas Röckmann[11], William T. Sturges[2], Colm Sweeney[4], Max Thomas[2], Elinor Tuffnell[2], and Felix Ploeger[1,12]

[1]Institute of Energy and Climate Research: Stratosphere, Jülich Research Centre, Jülich, 52428, Germany

[2]School of Environmental Sciences, University of East Anglia, Norwich, NR4 7TJ, United Kingdom

[3]Faculty of Science, University of East Anglia, Norwich Research Park, Norwich, NR4 7TJ, United Kingdom

[4]Cooperative Institute for Research in Environmental Sciences, University of Colorado-Boulder, Boulder, CO 80309, USA

[5]Global Monitoring Division, National Oceanic and Atmospheric Administration, Boulder, CO 80305-3337, USA

[6]Air Chemistry Division, Max Planck Institute for Chemistry, Mainz, 55128, Germany

[7]Center for Isotope Research, University of Groningen, Groningen, 9747 AG, The Netherlands

[8]Finnish Meteorological Institute, Sodankylä, 99600, Finland

[9]Department of Climate and Space Sciences and Engineering, University of Michigan, Ann Arbor, MI 48109-2143, USA

[10]Random Engineering Ltd., Felixstowe, IP11 9SL, United Kingdom

[11]Institute for Marine and Atmospheric Research Utrecht, Utrecht University, Utrecht, 3508 TA, The Netherlands

[12]Institute for Atmospheric and Environmental Research, University of Wuppertal, Wuppertal, 42119, Germany

*Correspondence to*: Johannes C. Laube (j.laube@fz-juelich.de)

**Abstract.** We present new observations of trace gases in the stratosphere based on a cost-effective sampling technique that can access much higher altitudes than aircraft. The further development of this method now provides detection of species with abundances in the parts per trillion (ppt) range and below. We obtain mixing ratios for six gases (CFC-11, CFC-12, HCFC-22, H-1211, H-1301, and $SF_6$), all of which are important for understanding stratospheric ozone depletion and circulation. After demonstrating the quality of the data through comparisons with ground-based records and aircraft-based observations we combine them with the latter to demonstrate its potential. We first compare it with results from a global model driven by three widely used meteorological reanalyses. Secondly, we focus on CFC-11 as recent evidence has indicated renewed atmospheric emissions of that species relevant on a global scale. Because the stratosphere represents the main sink region for CFC-11,

potential changes in stratospheric circulation and troposphere-stratosphere exchange fluxes have been identified as the largest source of uncertainty for the accurate quantification of such emissions. Our observations span over a decade (up until 2018) and therefore cover the period of the slowdown of CFC-11 global mixing ratio decreases measured at the Earth's surface. The spatial and temporal coverage of the observations is insufficient for a global quantitative analysis, but we do find some trends that are in contrast with expectations; indicating that the stratosphere may have contributed to the slower concentration decline

in recent years. Further investigating the reanalysis-driven model data we find that the dynamical changes in the stratosphere required to explain the apparent change in tropospheric CFC-11 emissions after 2013 are possible, but with a very high uncertainty range. This is partly caused by the high variability of mass flux from the stratosphere to the troposphere, especially at time scales of a few years, and partly by large differences between runs driven by different reanalysis products, none of which agree with our observations well enough for such a quantitative analysis.

**1 Introduction**

Many halogenated trace gases are strong greenhouse gases and/or are involved in the ongoing depletion of the ozone layer, therefore observations of these in the stratosphere are valuable. Moreover, measurements of some of these species allow to constrain changes in stratospheric circulation and transport across the tropopause. An analytical challenge is posed by the low abundances of many such gases, in combination with the low ambient pressures found in this region of the atmosphere. Another

challenge is the ability to reach the stratosphere as even the highest-flying research aircraft can only reach altitudes just above 20 km (Schauffler et al., 2003, von Hobe et al., 2013). This is modest considering that the stratosphere extends to around 50 km. Large high altitude balloons can reach altitudes of up to about 36 km (Engel et al., 2009, Ray et al., 2017) but due to the heavy payloads are increasingly difficult to fly given the risks for people living in landing areas and the cost or risk from lifting gases such as helium or hydrogen. Satellite (or aircraft) remote sensing plays an important role and can offer a global picture

for some gases (Stiller et al., 2008, Santee et al., 2013, Harrison et al., 2019), but measurement precisions and altitude resolution are often limited. They are also indirect observations and require continued validation through independent *in situ* methods. Generally, the mentioned platforms are rather expensive, ranging from costs on the order of € 10,000 per flight hour for aircraft, to satellite costs of millions of Euros. The relatively recently developed AirCore technique (Karion et al., 2010), with flight costs of below € 2,000 (depending on the setup) offers a cost-effective alternative. AirCores, which were named due to

similarities to ice cores extracted from glaciers, are based on the concept of flying a very long lightweight coiled piece of stainless steel tubing on a weather balloon. The tube is open on one end and therefore empties naturally upon ascent as ambient pressures decrease. During descent a full vertical profile of air is collected between the balloon's burst altitude (up to 36 km) and ground level. This technology has been demonstrated before, but for providing measurements of only a handful of higher abundance trace gases such as $CO_2$ and $CH_4$ (Karion et al., 2010, Membrive et al., 2017, Engel et al., 2017) and their isotopic

composition (Mrozek et al., 2016, Paul et al., 2016).

However, due to the limited amount of air that is captured by AirCores, no ozone-depleting substances (ODSs) have been investigated yet, as their abundances are well below one part per billion. The importance of such observations is however demonstrated by the following example. The recent work of Montzka et al. (2018) on renewed emissions of CFC-11 has received much attention since it indicates a substantial and ongoing breach of the global treaty designed to prevent the destruction of the stratospheric ozone layer: the Montreal Protocol on Substances that Deplete the Ozone Layer. According to their study, global CFC-11 emissions increased by $13 \pm 5$ Gg yr$^{-1}$ when comparing periods before and after 2012 with the caveat that up to 50 % of that derived emission change might actually be attributable to changes in stratospheric processes or dynamics. More recently, Rigby et al., (2019) found similar global increases of 11-17 Gg/year over 2014-2017 vs the 2008-2012 average, and also pinpointed a concurrent emissions increase source of $7.0 \pm 3.0$ Gg yr$^{-1}$ to eastern mainland China. However, they found no emission increases in other parts of the world covered by regular ground-based observations. This could mean that some of these emission increases have arisen in regions where no such measurements are available. An alternative explanation, i.e. the possibility of a sustained change to the amount of CFC-11 exchanged between the troposphere and the stratosphere as the driving mechanism for at least a part of the anomaly, has however not been ruled out so far.

## 2 Methods

Dry air mole fractions of halogenated trace gases were derived from air samples collected on board three different platforms: a passenger aircraft (CARIBIC, Brenninkmeijer et al., 2007) flying at altitudes of 8-13 km (11 flights, 2009-2016), a research aircraft (Leedham Elvidge et al., 2018) accessing higher altitudes of 9-21 km (M55 Geophysika, five campaigns, 2009-2017), and the first measurements of such gases with the relatively recently developed AirCore methodology (Karion et al., 2010, 8-30 km, 15 flights in Finland and the UK, 2016-2018). The aircraft data have partly been published before (Leedham Elvidge et al., 2018, Laube et al., 2013). The balloon-based AirCore technique was developed further mainly through the use of specially-designed tubing that maximises the amounts of air collected in the stratosphere, as well as through a novel subsampling technique that minimises the use of contamination-prone materials. The amount of retrievable stratospheric air is however still more than two orders of magnitude smaller than from aircraft-based sampling techniques. With laboratory analytical improvements compensating for this, the AirCore measurements show good precisions (ranging from 0.2 to 3.3 % compared with 0.4 to 1.1 % for aircraft samples) and excellent agreement with the aircraft data. The other important challenge for AirCore measurements of halocarbons is to ensure that the air is not contaminated throughout the entire sampling and sub-sampling process. Contaminations can arise from leakages and/or halocarbon-emitting materials (such as organic polymers) in the AirCore itself, in the $CO_2$-analyser system including the pump, or in the subsampling system. Importantly, for all compounds reported here mixing ratios in the stratosphere are much lower than in even remote tropospheric regions, let alone near sources of these gases. In addition, almost all of the contamination possibilities would affect the entire profile as an AirCore is essentially one air sample. This would become apparent in the correlations of the species with each other, which are very compact in the stratosphere. In the absence of such correlation breakdowns (See Figures 1, 2, and S1 to S4) we

therefore conclude that such contaminations are at undetectable levels in the data set presented here. More details can be found in Table 1 and the supplement.

All samples were processed with a previously described analytical system and methodology (Laube et al., 2010 and 2012,) using cryogenic extraction and pre-concentration, followed by gas chromatographic separation and detection with a high-sensitivity mass spectrometer. Trace gas measurements from this system as well as mean Ages of Air (AoAs, i.e. average stratospheric transit times, see section 3.1 for more details) calculated from these have been shown to compare very well with those of other internationally recognised measurements over several decades (Leedham Elvidge et al., 2018, Laube et al., 2013,

Trudinger et al., 2016).

Table 1. Comparison of average measurement uncertainties (derived as the average one standard deviation from repeated working standard or air sample measurements) of the research aircraft campaign in 2016, all AirCore flights, and some AirCore sample repeats for CFC-11, CFC-12, H-1211, H-1301, HCFC-22, and $SF_6$. For AirCore uncertainties the average working

standard uncertainty over three years was used as it is a) more representative of the entire measurement period and b) generally comparable or worse than precisions derived from sample repeats. AirCore-based precisions are generally slightly worse than those achieved with the larger aircraft-based samples, but still much smaller than mixing ratio gradients observed in the stratosphere.

| Trace Gas / Average precision [%] | Aircraft 2016 | AirCore 2016-18 Standards | AirCore 2017 Sample repeats |
|---|---|---|---|
| CFC-11 ($CFCl_3$) | 0.4 | 0.9 | 1.2 |
| CFC-12 ($CF_2Cl_2$) | 1.1 | 1.2 | 0.8 |
| H-1211 ($CF_2ClBr$) | 0.6 | 1.9 | 1.0 |
| H-1301 ($CF_3Br$) | 0.6 | 3.3 | 2.3 |
| HCFC-22 ($CHF_2Cl$) | 0.6 | 0.9 | 0.2 |
| $SF_6$ | 0.4 | 0.9 | 0.6 |


Stratospheric trends at AoA surfaces were derived by fitting second- and third-order polynomials (depending on whether an inflexion point was observed) to the respective correlations of mixing ratios and AoAs. The formulas of the polynomials were then used to interpolate onto the AoA surfaces (1, 2, 3, and/or 4 years, depending on which AoA range was covered) for each flight. To test the uncertainty of this method, the data for each flight was first quintupled by adding and subtracting the mixing

ratio and mean age uncertainties. This resulted, for each data point, in the average plus minimum and maximum value for both mixing ratio and AoA. Subsequently 5n (n being the number of data points available for each flight) random samples were

drawn (repeat draws possible) with a bootstrap algorithm (as in Volk et al., 1997 and Laube et al., 2013), and a second or third order polynomial again fitted. This procedure was repeated 500 times for each flight resulting in an average mixing ratio and an uncertainty range at each AoA surface. The derived mixing ratios were subsequently used to produce linear regressions over time, including a weighting by the inverse uncertainties of the individual CFC mixing ratios. The bootstrapping algorithm (500 repeat draws, repeat draws possible) was used again to ensure that the derived slope uncertainties were not underestimated and that individual high or low points did not bias the slope estimates.

Observation-based data were compared to model output from the Chemical Lagrangian Model of the Stratosphere (CLaMS), a Lagrangian chemical transport model with advective transport calculated from three-dimensional forward trajectories and an additional parameterisation for small-scale turbulent mixing (McKenna et al., 2002). Potential temperature is used as vertical coordinate throughout the stratosphere with vertical velocity estimated from the total diabatic heating rate. Further model details and the chemistry scheme used are described in Pommrich et al. (2014). For the simulations used in this study CLaMS was driven with horizontal winds and diabatic heating rates from three alternative meteorological reanalysis data sets: ERA-Interim (from European Centre for Medium-Range Weather Forecasts, ECMWF), JRA-55 (from Japan Meteorological Agency), and MERRA-2 (from NASA). For more information on methods, calibrations, and modelling as well as additional data please see the Supplementary Information.

## 3 Results and discussion

### 3.1 Observational data overview and comparisons

Our data are based on measurements of air samples collected in the upper troposphere and stratosphere of the northern hemisphere using aircraft and weather balloons between 2009 and 2018. Figure 1 shows the obtained mixing ratios alongside the northern hemispheric 'background' time series derived through the combination of observations at various ground-based stations within the National Oceanic and Atmospheric Administration Global Monitoring Division's Global Greenhouse Gas Reference Network (NOAA/GMD GGGRN). It is apparent that both the aircraft and the balloon data follow the ground-based trends quite well for all six gases. Slightly enhanced mixing ratios can often be observed in the vicinity of the tropopause (see also Figures S5 and S6), mostly due to recent influences from regional emissions (Kloss et al., 2014, Leedham Elvidge et al., 2015, Oram et al., 2017). This is especially pronounced in the research aircraft data from 2017, which belong to a campaign (Höpfner et al., 2019) exploring the atmospheric composition above the polluted Asian Monsoon region (Randel et al., 2010, Vogel et al., 2019). It is however worth noting that most species' enhancements are not significantly higher than the combined measurement uncertainties, which demonstrates the importance of the consistency of the data sets and therefore the quality of the stratospheric record. Figure 1 also illustrates the much improved temporal density that AirCore observations have provided from 2016 onwards (in comparison to aircraft campaigns), especially at altitudes above 15 km which are out of the reach of all but a few research aircraft.

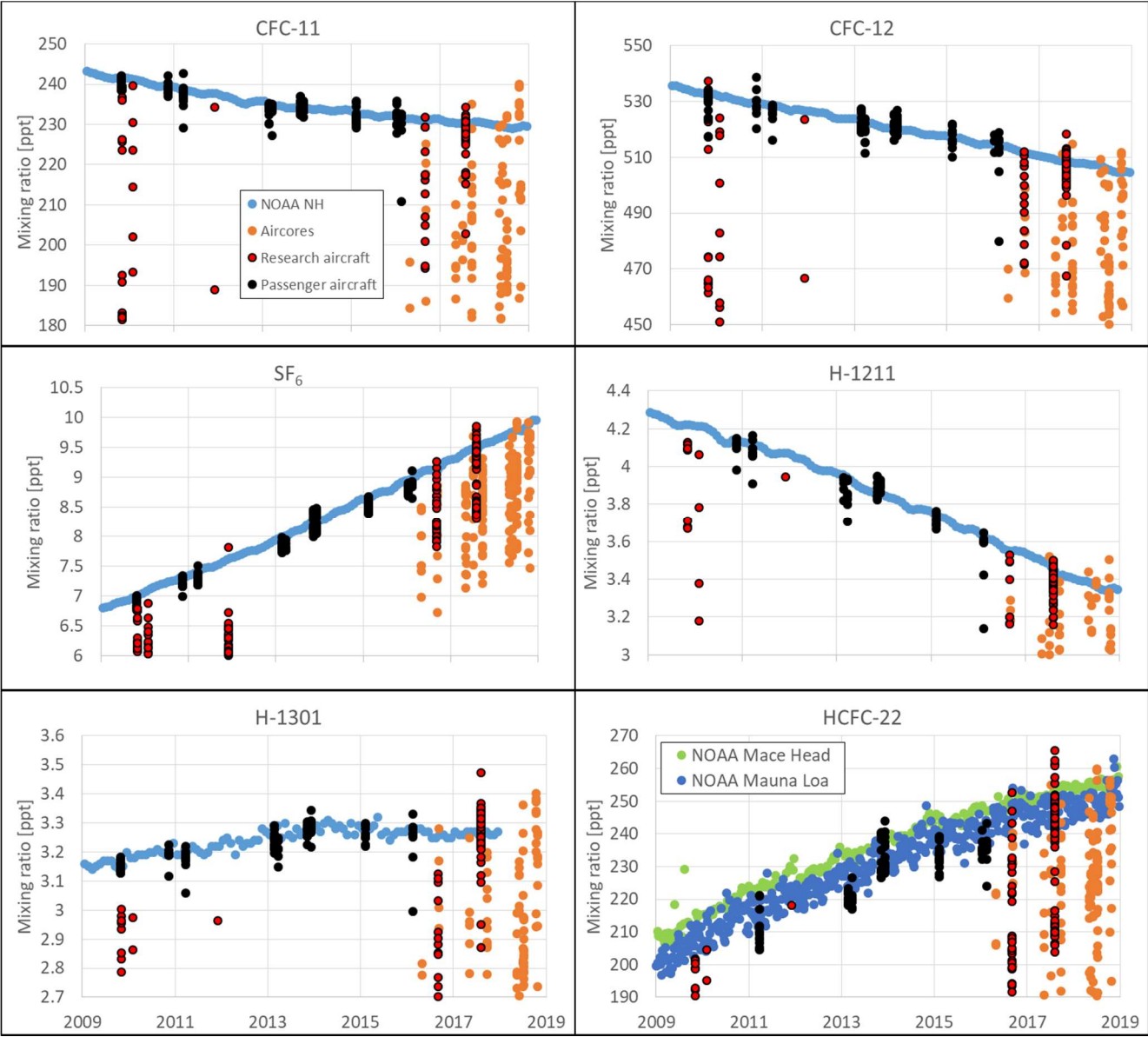

**Figure 1. Aircraft- and balloon-based mixing ratios of six halogenated trace gases in the upper troposphere and stratosphere as compared to the NOAA/GMD ground-based northern hemispheric GGGRN time series (https://www.esrl.noaa.gov/gmd/). HCFC-22 has a significant sink process in the troposphere and therefore exhibits stronger inner-hemispheric gradients. To illustrate that, we compare against the mid-latitude station at Mace Head, Ireland and the subtropical station at Mauna Loa, Hawaii. Lower mixing ratios generally represent higher altitudes. For all gases except SF$_6$ some higher altitude data are not shown to better demonstrate the good comparability of near-tropopause data to the NOAA time series. The complete corresponding data including uncertainties can be found in the supplement (see also Figures S1 to S4).**

In the stratosphere trace gases typically exhibit compact interspecies correlations (Schauffler et al., 2003, Volk et al, 1997), and some gases (such as $SF_6$) can be utilised to derive average stratospheric transit times, which are more commonly known as mean Ages of Air (AoA, Engel et al., 2009, Ray et al., 2017, Stiller et al., 2008, Leedham Elvidge et al., 2018). The
correlations between CFC-11 and CFC-12 as well as between CFC-11 and AoA derived from observations (see S1.2 for details) are shown in Figure 2. Two things are apparent: firstly, this again demonstrates the consistency and quality of our data as similar correlations are observed for both aircraft- and AirCore-based mixing ratios over the entire range. Secondly, the correlations have not undergone a large shift in the last ten years. Correlations between trace gases are often driven by changes in tropospheric trends, as tropospheric air keeps "feeding" the stratosphere. A large shift in these correlations would therefore
not be expected as both CFC-11 and CFC-12 have experienced relatively small negative tropospheric trends in recent years (Montzka et al., 2018, Rigby et al., 2019). However, there are other factors that can change the correlations, namely changes in stratospheric chemistry and transport. The CFC-11-AoA correlation in particular would be affected if e.g. the main transport pathways and or times (AoAs) inside the stratosphere had changed. This possibility is investigated further below.

### 3.2 Comparisons with model data using different reanalyses

We first focus on a comparison of model simulations with the aircraft and AirCore data. Also shown in Figure 2 are data from simulations with the Chemical Lagrangian Model of the Stratosphere (CLaMS, McKenna et al., 2002, Pommrich et al., 2014). The latter was driven alternatively by three commonly used meteorological reanalyses, i.e. ERA-Interim, JRA-55, and MERRA-2 (Dee et al., 2011, Kobayashi et al., 2015, Gelaro et al., 2017). These newest available meteorological reanalysis datasets provide the best guess of the current state of the atmosphere. We use the differences between them to quantify the
uncertainty in our knowledge of the stratospheric circulation and its changes. The model was sampled at coordinates and times coinciding with those of the observations. Results from all three runs are similar to those from observations in the case of the correlation of CFC-11 with CFC-12. The CFC-11-AoA correlation in Figure 2 is a measure of the speed of the main stratospheric overturning circulation as it reflects, in an integrated way, the speed and pathway of trace gas transport through the stratosphere. Here, the model data for both ERA-Interim and JRA-55 remain close to the observed values throughout the
range. The MERRA-2-based data does however stand out producing higher AoAs at similar stratospheric CFC-11 mixing ratios and an increasing discrepancy with increasing AoA. As noted by Ploeger et al. (2019) the MERRA-2 reanalysis has a slower stratospheric circulation, and our observational evidence strongly indicates that it is indeed too slow. This is a consistent feature, which is also apparent when comparing with MERRA-2-based data from before 2016 (not shown in Figure 2). The details of the causing mechanisms could be complex and are beyond the scope of this work.

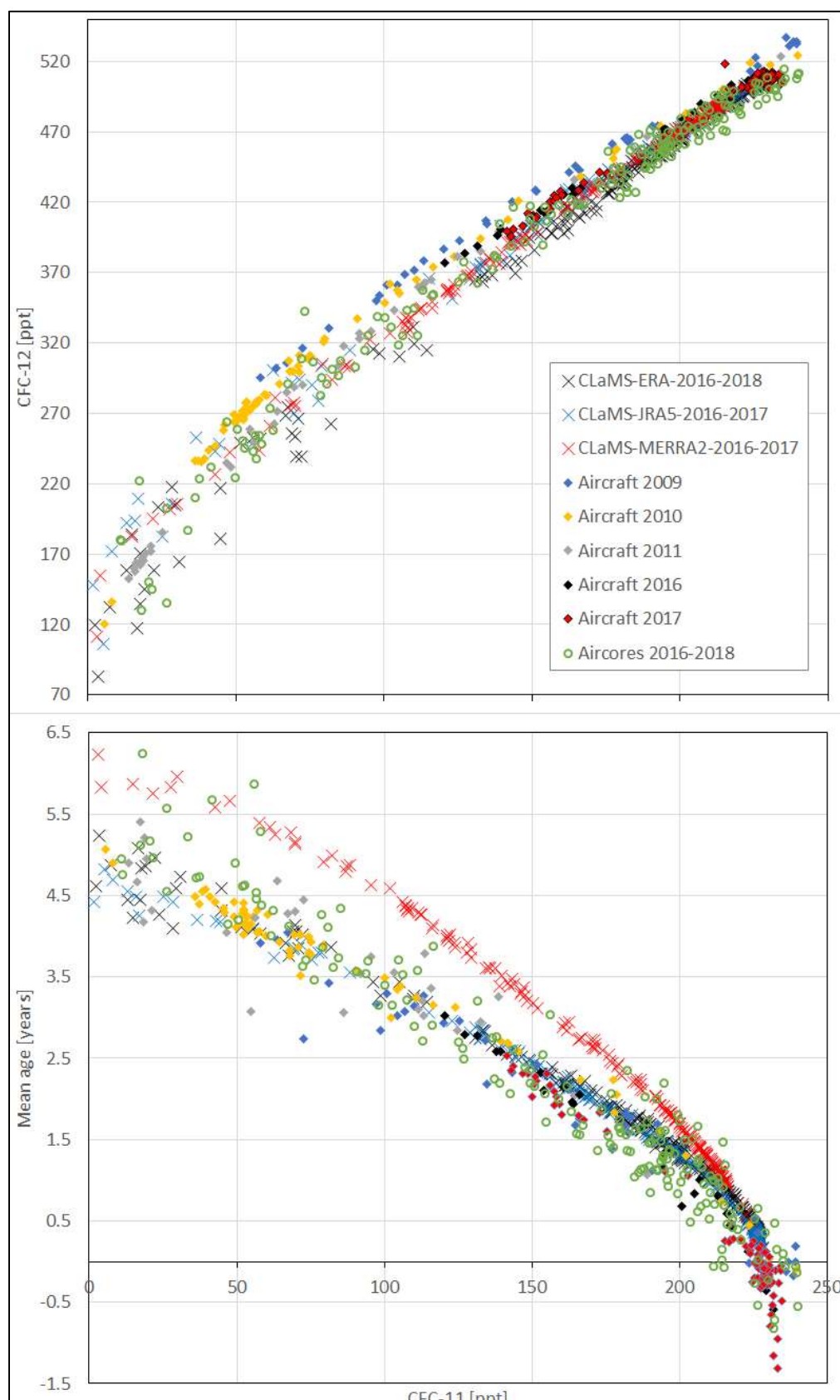


**Figure 2. Stratospheric CFC-12 mixing ratios and the mean age of air (AoA) as a function of CFC-11 mixing ratios, as observed in air samples collected by research aircraft (diamonds) and AirCores (circles). Crosses denote the values obtained from the CLaMS model sampled at the same times and coordinates as the observations, but for better visibility only from 2016 onwards. The CLaMS model was run using three different meteorological reanalysis packages: ERA-Interim (black), JRA-55 (blue), and MERRA-2 (red).**

## 3.3 Long-term trends of trace gases in the stratosphere

Focusing on the details of the correlations in Figure 2, we investigate whether there are indications here that might partly explain the recent changes in the tropospheric trend of CFC-11. Most air enters the stratosphere in the tropics and is then transported poleward. CFC-11 and CFC-12 molecules are mostly destroyed in the tropical stratosphere (Douglass et al., 2008). Transport of the remainder of these gases to the poles is much slower than in the troposphere, and takes several years (Kida, 1983, Schmidt and Khedim, 1991) as is reflected in the CFC-11-AoA correlation in Figure 2. In case of an acceleration of parts of the circulation, for which there have been observational indications (Bönisch et al., 2011, Stiller et al., 2012), that correlation should therefore shift. We consequently fitted the CFC-11-AoA correlation with a second or third order polynomial for each individual research aircraft and balloon flight and calculated the mixing ratio of CFC-11 after having spent, on average, one, two, three and four years in the stratosphere. Figure 3 shows examples of the trends at the four residence times from 2009 to 2018, and the full data can be found in the supplement.

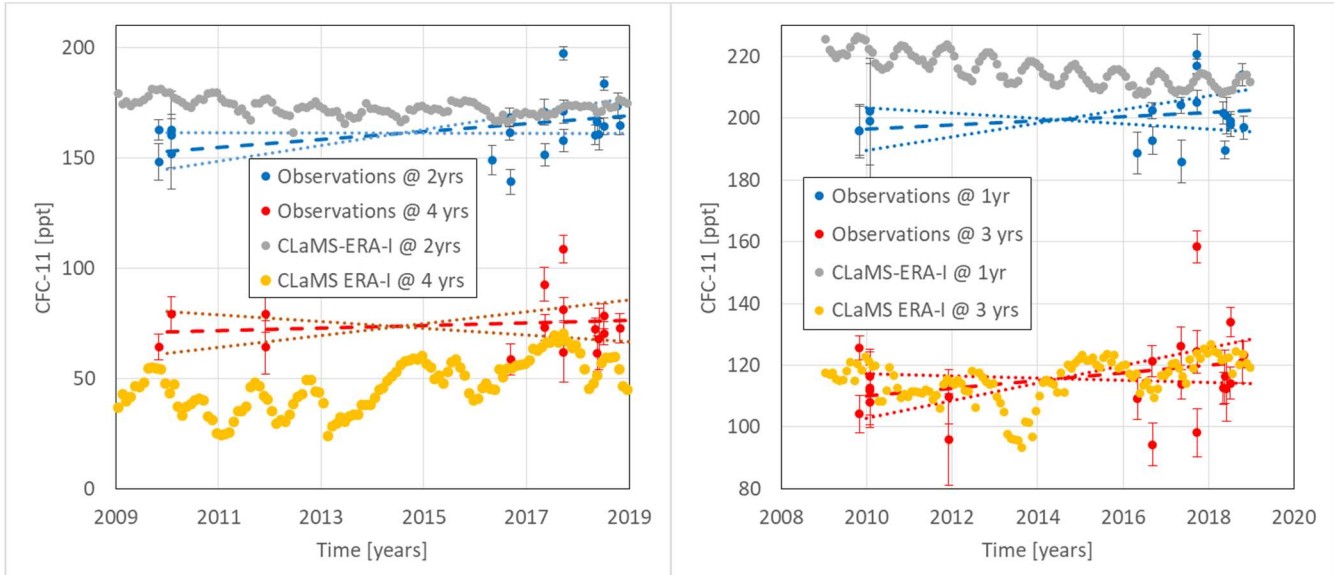

**Figure 3. The left panel shows a comparison of observation-based CFC-11 mixing ratio trends at mean ages of air of two (blue) and four (red) years with those from the CLaMS model run driven by the ERA-Interim reanalysis (grey and yellow) in the northern hemisphere stratosphere. The latter have been derived as averages between 30 and 90 °N. The dashed and dotted lines correspond to regression lines (weighted by their one σ standard error for observations) and an illustration of their two σ uncertainties over the**

**time periods displayed. The right panel shows the same comparison, but at mean ages of one and three years. The numerical values can be found in Table 2.**

While there is substantial variability of mixing ratios at these AoA surfaces over time, we do find a positive trend (increases from 3 to 10 %) from 2009 to 2018 for all observation-based (aircraft and AirCore) estimates. The trends at an AoA of one and four years are not significantly positive, but the ones at two and three years are; within 2.0 and 1.6 standard deviations of the slope uncertainties respectively (Figure 3, Table 2). These stratospheric trends contrast the tropospheric trend of CFC-11 which has been negative throughout that period (~-6% in total, Figure 1). While there is a certain lag time for air to reach our stratospheric observation points (i.e. 1, 2, 3, and 4 years on average), CFC-11 had been decreasing nearly linearly in the troposphere since the late 1990s. In turn this implies that changes in stratospheric circulation may indeed have played a substantial role in the recent changes to the tropospheric trend of CFC-11 as previously suspected (Montzka et al., 2018). The causes are not explicable with an integrated quantity such as AoA as the underlying distribution of stratospheric transit times cannot currently be inferred from trace gas observations. However, it should be noted that the limited temporal and spatial coverage of the observation-based and especially the gap between 2011 and 2016 introduces an additional uncertainty that is only partly reflected in the derived slope uncertainties.

Table 2. Temporal trends and their two σ uncertainties of CFC-11, CFC-12, HCFC-22, and H-1211 mixing ratios at AoAs of one, two, three, and four years. These slopes correspond to an uncertainty-weighted regression line fitted to the data in Figures 3, and S8-S12, with two exceptions: 1) The data from the Asian Monsoon campaign in 2017 was excluded as this is region is not representative of NH stratospheric air and 2) all data at mean ages above 3.5 years from winter campaigns in high latitudes was also excluded as it might contain polar vortex air, which is equally unrepresentative. Model-based slopes were derived over the same period as observational data (2009.8-2018.8), except for JRA-55 and MERRA-2, where data was only available until the end of 2017.

| CFC-11/AoA | 1 year | 2 years | 3 years | 4 years |
|---|---|---|---|---|
| Slope obs (ppt/year) | 0.69 | 1.77 | 1.25 | 0.59 |
| Uncertainty (ppt/year) | 1.54 | 1.81 | 1.60 | 2.12 |
| Trend (%/decade) | 3.2 | 10.4 | 10.2 | 7.4 |
| Slope ERA-I | -1.35 | -0.50 | 1.15 | 3.09 |
| Uncertainty | 0.22 | 0.24 | 0.47 | 0.61 |
| Slope JRA-55 | -1.56 | -1.38 | -0.08 | 1.73 |
| Uncertainty | 0.21 | 0.20 | 0.27 | 0.62 |
| Slope MERRA-2 | -1.69 | -1.51 | -1.20 | -0.55 |
| Uncertainty | 0.18 | 0.23 | 0.23 | 0.30 |

**CFC-12**

| Slope obs (ppt/year) | -1.96 | -0.45 | -0.38 | -1.33 |
|---|---|---|---|---|
| Uncertainty (ppt/year) | 1.90 | 2.20 | 2.52 | 5.36 |
| Trend (%/decade) | -3.6 | -0.95 | -0.93 | -3.9 |
| Slope ERA-I | -3.09 | -2.52 | -1.52 | 3.21 |
| Uncertainty | 0.37 | 0.48 | 0.62 | 1.20 |
| Slope JRA-55 | -3.17 | -3.09 | -1.37 | 2.39 |
| Uncertainty | 0.28 | 0.41 | 0.60 | 1.08 |
| Slope MERRA-2 | -3.26 | -3.17 | -3.02 | -2.40 |
| Uncertainty | 0.24 | 0.39 | 0.51 | 0.76 |

**HCFC-22**

| Slope obs (ppt/year) | 6.15 | 6.16 | 5.98 | 5.67 |
|---|---|---|---|---|
| Uncertainty (ppt/year) | 0.15 | 0.14 | 0.14 | 0.18 |
| Trend (%/decade) | 30.5 | 33.4 | 36.0 | 38.2 |

**H-1211**

| Slope obs (ppt/year) | -0.031 | 0.000 | 0.013 | 0.013 |
|---|---|---|---|---|
| Uncertainty (ppt/year) | 0.008 | 0.008 | 0.007 | 0.009 |
| Trend (%/decade) | -9.0 | 0.2 | 9.1 | 22.4 |

230   For the other three gases that have sufficient measurement precisions for such an analysis (i.e. CFC-12, H-1211, and HCFC-22), we also find a picture that does not agree well with their tropospheric trends (Table 2). Both CFC-12 and H-1211 have been in decline in the troposphere since the mid-2000s and decreased by ~6 and ~20 % respectively between late 2009 and late 2018 (Figure 1); whereas tropospheric HCFC-22 mixing ratios have increased monotonously (and by ~25 % during our observation period) since its appearance in the atmosphere several decades ago, albeit with a recent slowdown. In contrast, in

235   the stratosphere, we find that CFC-12 decreased at all mean age surfaces, but not as much as in the troposphere (-0.9 to -4 %), HCFC-22 increased disproportionally by 30 to 38 %, and H-1211 decreased, but only at a mean age of one year (-9 %), while no significant change occurred at 2 years, and 9 to 22 % increases were observed at 3 and 4 years. For the latter three gases this unexpected behaviour could be partly related to changes in tropospheric trends in the period leading up to 2009, as a significant part of the air at certain mean age levels is much older than the mean age itself. However, these effects should

240   subside over the decade that our observations span, especially for H-1211, which is the shortest-lived gas of the four. In

addition, CFC-11 should not be affected as it has been decreasing for much longer. The underlying mechanisms are likely complex.

The only straight-forward possibility to generate positive CFC-11 trends in the stratosphere between 2009 and 2018 would be an increase in the air fractions that have younger and older residence times than the inferred mean age. Such a two-fold increase would maintain the same AoA, but would influence the mixing ratios observed at the AoA surfaces in different ways. If the increased older air fraction had been in the stratosphere for long enough, it would have already lost virtually all of its content of shorter-lived gases (H-1211 and CFC-11). However, if this older air fraction at the same time would be in an AoA range where the longer-lived gases (CFC-12 and HCFC-22) are still present in significant amounts, then an increase in its share should lead to a decrease in CFC-12 and HCFC-22 mixing ratios (but less so for the latter as it is much longer-lived in the stratosphere). To balance this increase in the older air fraction and maintain a constant mean age, the younger fraction of the AoA spectrum would also need to have an increased share. Younger air generally contains higher mixing ratios of all four gases – and disproportionally so for HCFC-22 as its tropospheric mixing ratios continue to increase. If the increases in the two fractions of the AoA spectrum would be in the right AoA range, the overall effect would then be an increase of mixing ratios of CFC-11, H-1211, and HCFC-22 over time at a given AoA surface, accompanied by a decrease in CFC-12 mixing ratios. This would then be entirely consistent with the changes we observed at almost all AoA levels between 2009 and 2018. Therefore such a change to the stratospheric transit time distributions could be considered as the simplest case that would qualitatively explain our observations.

The aforementioned possibility to at least partly explain such trends could include an acceleration of air mass transport through the lower tropical stratosphere (i.e. below the main sink region of CFC-11) as for example CLaMS-ERA-Interim qualitatively shows over the relevant period (Figure S15). However, when compared with ERA-Interim-based model data at the same transport times (Figure 3), the model results show a different CFC-11 trend in the lower stratosphere. In fact the model- and observation-based trends at mean ages of one and two years do not agree within two standard deviations. This discrepancy is likely related to a known problem with ERA-Interim, which generally overestimates the speed of the circulation in that lower stratospheric region (Dee et al., 2011, Ploeger et al., 2012). At larger mean ages we find better agreement between the observations and the model with the model data even reproducing the observed insignificant trend. JRA-55-based model trends are very similar to those from the ERA-Interim-based analysis whereas the MERRA-2 reanalysis shows larger differences to observations, both in terms of mixing ratios and trends (Table 2, Figures 3 and S8-S12). The generally limited comparability of model and observations sheds some light on the ability of current reanalysis products to quantify structural changes in stratospheric circulation patterns.

## 3.4 Mass flux estimates of CFC-11

Nevertheless we use the reanalysis-driven model data as the best available means to derive the downward mass flux of CFC-11 through the extra-tropical tropopause, i.e. the quantity describing how much CFC-11 is transported back to the troposphere.

Comparing the three simulations driven with three different reanalyses provides an estimate of uncertainty due to representations of stratospheric circulation changes. A temporal increase of the stratosphere-to-troposphere mass flux could cause changes to the tropospheric trend of CFC-11, which would look like renewed emissions. Such a flux increase could be consistent with the observed increases in CFC-11 mixing ratios on AoA surfaces (Section 3.3) if accompanied by an increased fraction of air entering the stratosphere without passing through the main CFC-11 sink region in the lower tropical stratosphere (and instead entering e.g. through the Asian Summer Monsoon).

The NOAA/GMD tropospheric time series of CFC-11 serves as the boundary condition to the model, and consequently in the absence of stratospheric changes the temporal trend of the mass flux should be similarly negative and of a similar magnitude. The model generally reflects this reasoning over longer time periods as can be seen in Figure 4. We then follow the approach of Montzka et al. (2018) to investigate whether the changes to the tropospheric trend around 2013 might partly be caused by more CFC-11 being transported back into the troposphere. For that purpose we split the data into two periods, before and after 2013. Independent on which definition of the tropopause is being used (see supplement for details), we find an increase in the mass flux of around 37 Gg/year after 2013 for CLaMS-ERA-Interim. This would explain 270 % of the observed slowdown of CFC-11 mixing ratio decreases after 2013 when comparing to the $13 \pm 5$ Gg/year emission increase inferred by Montzka et al. (2018). At first glance, this very high stratospheric contribution is not consistent with the findings of both Montzka et al (2018) and Rigby et al. (2019), who estimated 40-60 % of the slowdown to belong to renewed emissions. However, the global stratosphere-to-troposphere mass flux is very large compared to the amount of unexplained emissions and a direct quantitative comparison is not possible, as explained in the following. When repeating the same model run, but with an artificial tropospheric CFC-11 trend that continues to decrease linearly after 2013 the mass flux remains very similar to the reference simulation (difference of < 0.6 Gg/year). This translates into a minor influence of recent tropospheric trend changes on these stratospheric fluxes therefore confirming that this signal is indeed driven by stratospheric changes in the ERA-Interim world. However, this pronounced turnaround in 2013 is not a consistent feature for all three reanalyses, as the JRA-55 run, despite producing such a similar picture in the correlation comparisons (Figure 2), in fact shows a further decrease of 0.4 Gg/year (equivalent to -3 % of the new emissions signal) after 2013. The main reason for that discrepancy is that, as opposed to ERA-Interim, JRA-55 does not show a substantial change to the mass flux around 2013. Coming back to the pre- and post-2013 analysis, CLaMS-MERRA-2 results are in between the other two with 18.2 Gg/year (135 %), but have the least credibility as demonstrated by the poor comparability with observations. The main issue connected with such an analysis is illustrated in Figure 4. With annual changes of up to 21 % the variability of the CFC-11 mass flux from the stratosphere to the troposphere is an order of magnitude higher than the 2013 change of 2-5 % that we are trying to quantify. Some of that mass flux variability occurs over several years, which severely limits the capability of quantitatively determining trend changes between an eleven- and a five-year period. It should however be reemphasised that a mass flux trend analysis over longer periods would be expected to work better and this is indeed what we find for ERA-Interim and JRA-55. Between 2002 and 2017 the CFC-11 flux from a linear regression of the model output driven by these two reanalyses decreases by 10.5 and 13.1 % respectively, which is comparable to the ~11 % tropospheric decrease over the same period. MERRA-2 again produces an outlier with only

a 3.2 % decrease during those 16 years. The recent findings of Ray et al. (2020) of the QBO significantly modulating the variability of long-lived trace gases at the surface are qualitatively consistent with our findings for both shorter and longer periods. However, a quantification of this modulation is currently limited by the uncertainties connected to the meteorological reanalyses in the stratosphere. As shown in Figure 4 the mass fluxes from the three CLaMS-reanalysis runs show some covariation on QBO time scales, but at the same time also some significant differences which include offsets, long-term trends, the magnitude of the variations, and the timing of changes.


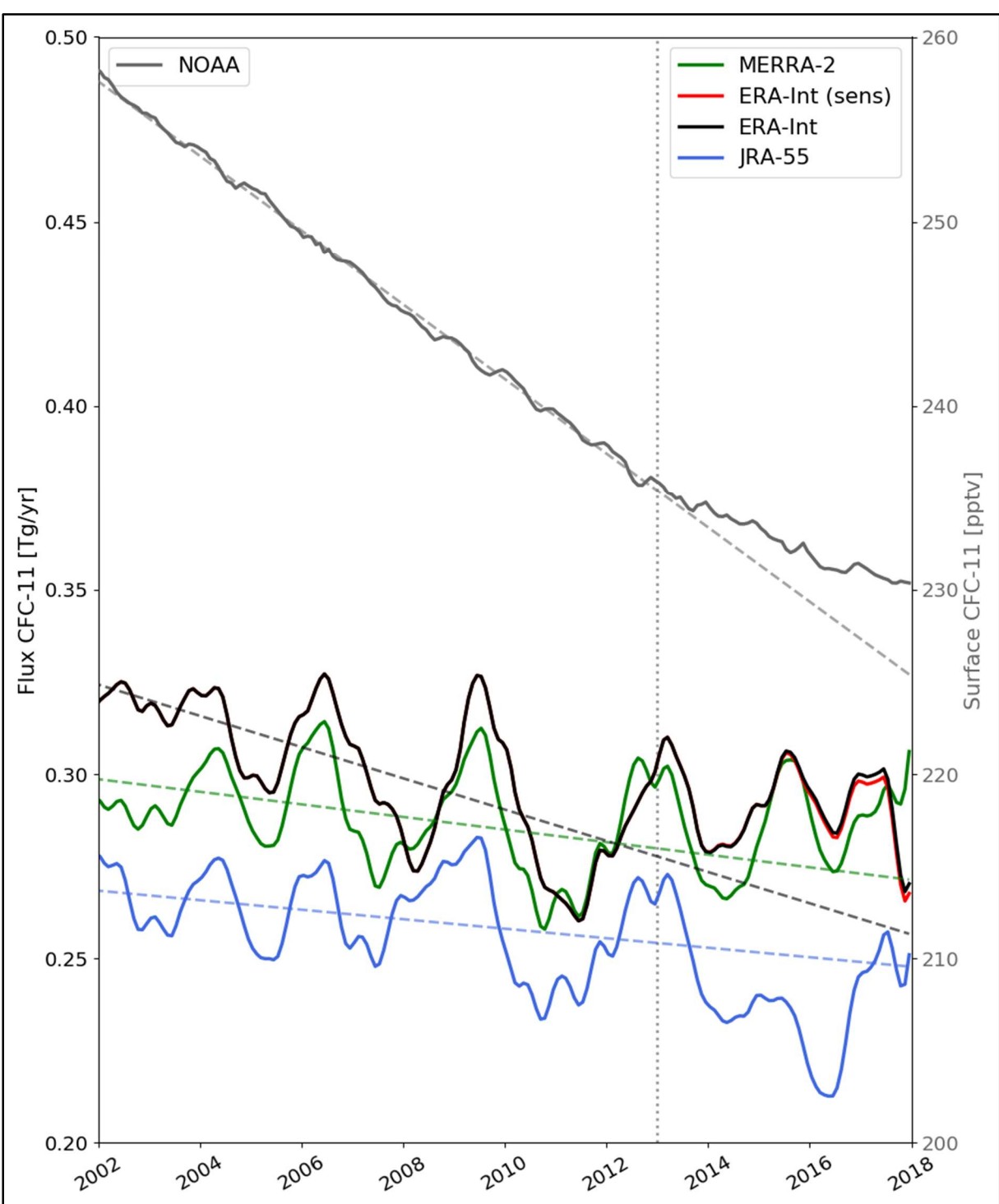

**Figure 4. The annually averaged stratosphere-to-troposphere mass flux of CFC-11 through the tropopause between 2002 and 2018 for CLaMS model runs driven by MERRA-2 (green), JRA-55 (blue) and ERA-Interim (black) reanalyses including a linear**

**regression for the period until 2013 (dashed). The red line originates from an ERA-Interim sensitivity run for which tropospheric CFC-11 was forced to continue to decrease at the same rate as before 2013. Shown in grey and on the right hand y-axis are the two corresponding time series of tropospheric CFC-11 mixing ratios (i.e. the real one, solid, and the one with the forced decrease, dashed).**
**The annual average has been calculated by applying a 12 month running mean to the time series.**

## 4 Conclusions

To summarise, we present new observations of six halogenated trace gases in the stratosphere obtained from applying a further developed AirCore technology. These observations are consistent with ground-based measurements of the same species at remote locations. They compare well to aircraft-based observations, have good precisions, and offer a viable, low-cost method
for directly observing ozone-depleting gases and circulation tracers in the stratosphere at enhanced temporal and spatial resolution. The derived mixing ratios and mean stratospheric residence times both from aircraft and AirCore data enable the assessment of the performance of the three most modern currently available meteorological reanalysis packages. The ERA-Interim- and JRA-55-derived model data compare better whereas the MERRA-2-based data exhibits distinctly slower transport through most of the region covered here.

From a further analysis of the observational data at certain stratospheric transport times we also find insignificant to positive trends (within one standard deviation) of CFC-11 mixing ratios in the lower stratosphere between 2009 and 2018 ranging from 3 to 10 %. This is surprising and in contrast to expectation from the tropospheric abundances, which have been decreasing by about 6 % over that period. Similarly derived trends for CFC-12, HCFC-22, and H-1211 are also not in good agreement with their corresponding tropospheric trends. In a qualitative sense, and keeping in mind the regional nature of these measurements
and the uncertainties related to the calculation of stratospheric transport times, this would point towards increasing mass fluxes of CFC-11 being transported back to the troposphere. Our observations therefore do support the hypothesis of new emissions being lower than expected from tropospheric trends alone. More generally this is evidence towards a significant and time-dependent role of the stratosphere in the modulation of tropospheric trends of trace gases. However, any further quantification of the stratospheric part of the CFC-11 story is prevented firstly by the non-global and intermittent nature of sufficiently precise
observations as well as their limited comparability to model/reanalysis results; secondly by the variability of the CFC-11 stratosphere-to-troposphere mass flux influenced by e.g. QBO, ENSO, volcanic eruptions and also stratospheric transport changes as indicated by the observed trace gas trends on AoA surfaces; and thirdly by the large differences between results from different current meteorological reanalyses; with the quality of the latter currently being the main limitation to refining such calculations.

Finally, our observations span ten years, which is a short time in comparison to the long-term climate-change driven stratospheric circulation changes expected from global models, which are on the order of decades (Polvani et al., 2018). Our data however demonstrate the capabilities of the AirCore observations to increase data coverage and better constrain such changes on various time scales.

*Data availability.* Observational data are included in the Supplement and the CLaMS model data may be requested from the corresponding author.

*Author contributions.* J. C. L. conducted the analysis of the overall data set, participated in several campaigns, carried out some of the measurements, and led the writing of the manuscript. E. C. L. E., B. B., H. C., E. S. D., P. H., R. K., A. J. H., A. L., S.

R., C. S., M. T., E. T., and W. T. S. contributed to the design of the AirCore and subsampling equipment and the various balloon campaigns with E. C. L. E., E. S. D., and E. T. also involved in the halocarbon measurements and data analysis. C. A. M. B. and D. E. O. were responsible for the CARIBIC and T. R. for the Geophysika aircraft measurement and sampling equipment and related discussions. S. A. M. provided NOAA NH time series and useful respective insights, whereas J.-U. G. and F. P. led the modelling analysis. All authors contributed to the writing process of the manuscript scientific discussions

surrounding that.

*Competing interests.* The authors declare that they have no competing interests.

*Acknowledgements.* This work was funded by the ERC project EXC3ITE (EXC3ITE-678904-ERC-2015-STG). Johannes C.

Laube also received funding from the UK Natural Environment Research Council (Research Fellowship NE/I021918/1) and David E. Oram from the National Centre for Atmospheric Science. We gratefully acknowledge the computing time for the CLaMS simulations granted on the supercomputer JURECA at Jülich Supercomputing Centre (JSC) under the VSR project ID JICG11. Felix Ploeger was funded by the Helmholtz Association (Helmholtz Young Investigators Group A–SPECi, grant number VH-NG-1128). Karina Adcock was funded by the UK Natural Environment Research Council through the EnvEast

Doctoral Training Partnership (grant number NE/L002582/1). We thank all who helped with the balloon launches in Finland and the UK, the numerous NOAA station personnel and site scientists for sample flask collection and measurement, Michel Bolder for collecting the Geophysica air samples and acknowledge the work of the Geophysica aircraft team. Related funding came from ESA (PremierEx and FRM4GHG projects), the Forschungszentrum Jülich, the European Commission (FP7 projects RECONCILE-226365-FP7-ENV-2008-1 and StratoClim-603557-FP7-ENV-2013-two-stage, and H2020 project RINGO) and

the Dutch Science Foundation (NWO; grant number 865.07.001). We further thank Paul Konopka for carrying out some of the CLaMS simulations used here, Jörn Ungermann for help with code translations, and Rolf Müller for useful discussions.

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
