# Peer review of "Investigating stratospheric changes between 2009 and 2018 with trace gas data from aircraft, AirCores, and a global model focusing on CFC-11"

_Atmospheric Chemistry and Physics, 2020_

## Referee Comment (RC1) · Anonymous Referee #1 · 8 Apr 2020

Firstly, my sincere apologies that this review is arriving so late!

This paper presents novel AirCore measurements of several halocarbons in the stratosphere, and analyses trends in stratospheric trace gases, particularly focusing on CFC-11. The paper is interesting, timely, and the measurement technique shows great promise. I think the paper should be published in ACP, following some (mostly minor) changes.

General comments

In general, I think there is quite a lot of material in the supplement that should be in the main text. Particularly given that one of the main purposes of this paper is to

introduce a new measurement technique, I would encourage the authors to move the experimental details to the main paper, and some of the validation. I think that Tables S2 and S3 should be in the main text, as S2 demonstrates the repeatability of this new technique (which is referred to as a key conclusion of the paper), and the results of S3 are discussed at some length in the main text. Similarly, the brief description of the model setup could also be in the main text, as it is important for readers not familiar with this particular model.

Specific comments

The title needs to mention that changes in halocarbons are being referred to.

L34 – 35: "required" is used twice in the same sentence

L40: this first line is too vague. It doesn't really say anything. I suggest starting with the current second line.

L42: "low to very low" could just be replaced by "low"

L45: "upper limit is around 50km"? Is this suggesting that the stratosphere extends to 50km? Not clear as written.

Section 2 heading: "Methods" are a sub-section of "Results"? I suggest separating these two sections.

L75 and elsewhere: I'm not 100% sure about the grammar here, but I would have thought that a colon followed by a list separated by commas would be more appropriate here (after "platforms")

L80 – 83: As mentioned above, I think this needs much more explanation, and I would bring in a lot of the supplementary information.

L84: "good precision" needs quantifying. Perhaps compare to the precision of other measurement techniques and say whether it is comparable or not (avoid subjective terms like "good").

L88: AoA is defined later on (L125). Needs to be introduced here. Furthermore, I would suggest a brief paragraph here explaining the concept, and calculation method.

L90: I found this description too hard to follow. When were second or third order polynomials used? What exactly was being fitted? What is the purpose of "quintupling" the data, etc.? I think you need to walk the reader through this more gently. A figure may help to demonstrate the technique.

L134: what does "times inside the stratosphere" mean? Do you mean AoA?

L174: Given that this line is at the start of a paragraph, I think you need to re-state what each part of the sentence refers to. I.e. a positive trend in what? Explain "all observation based cases".

Figure 3: The conclusions drawn from this analysis, namely that there has been no trend, or a positive trend, seems highly dependent on a small number of points in 2009 − 2011. Can the authors comment on this? Are we really confident that the observations can provide strong conclusions about the overall trend, given that there are large periods of these time series with no data (i.e. nothing between ∼2011 and 2016)? Some softening of the discussion of the observation-based trend would seem warranted to me.

L175: as stated above, having Table S3 in the main text seems important to understand this sentence.

L187: For clarity, I suggest adding: "In contrast, in the stratosphere, we find. . ."

L195 − 204: I must admit to finding it very challenging to follow this explanation, despite reading over it three or four times. I realise it's a complicated business, but I suggest that the authors re-word, or, better, provide a schematic outlining their argument.

L197: I suggest: ". . . HCFC-22, which is much longer lived in the stratosphere" (it's shorter lived overall).

Section 2.4: I'm sure this highlights my ignorance, but doesn't it seem counterintuitive to have an increased mass flux from the stratosphere to the troposphere at the same time as an increase in stratospheric CFC-11? Can you add a couple of sentences explaining why this would be?

L230: "ratio", rather than "ratios"

Figure 4 caption: Specify which direction the flux is in (I think it strat to trop?).

---

## Referee Comment (RC2) · Anonymous Referee #2 · 27 Apr 2020

This paper is very well written and well referenced. It makes important contributions to; 1) evaluating comparisons of halocarbon measurements from different sampling platforms over a 10 year period, 2) the use of halocarbon measurements to evaluate stratospheric dynamics in a global model using different meteorological reanalyses, and 3) exploring the use of model derived atmospheric dynamics to examine reasons for the reduction in CFC-11 global mixing ratio decreases.

Overall this paper is high quality and should be accepted after addressing the few suggestions noted below.

Line 41: The phrase "allow to gauge" is awkward and should be revised.

[Figure]

Line 88: This is the first use of AoAs and it should be spelled out.

Line 90: Polynomial fit functions of what? I suggest adding in "of AoA vs mixing ratio of each species" or something like that.

Line 93: The bootstrap method isn't explained fully in Laube et al., 2013 so I suggest including Volk et al., 1997 with the Laube et al., 2013 reference here. A bit more explanation would also be helpful.

Line 108: I suggest including a reference to Figures S5 and S6 here.

Line 112: I suggest referencing Table S2 for the measurement uncertainties.

Lines 114-115: The authors state here that Figure 1 illustrates improved temporal density from 2016, which it does, but goes on to say especially at altitudes above 15 km. There is no indication of altitude in Figure 1. As the authors mention, the lower values are from higher altitudes, but there is nothing that says what those altitudes are, and in fact, the values in Figure 1 are not from the highest altitudes. . .see next comment re Figure 1.

Figure 1: The mixing ratios for all compounds do not reflect the full range of the combined measurements. The full range for CFC-11 and CFC-12 are seen in Figure 2 and for the other compounds in Figures S1-S4. Please indicate in Figure 1 why this is the case.

Line 146: The authors state the MERRA-2 based data stands out producing higher transport times at similar stratospheric CFC-11 mixing ratios. It should read "higher mean ages", rather than higher transport times. As the authors say in line 148, this results from slower transport times.

Line 176: I suggest including Figure S7 in the main text so the readers can see all four years rather than going to the Supplementary section for 2 of the 4 years.

Line 184: Please reference Figures S8-S13 for this discussion.

S12 and S13 Figure captions should be MERRA-2 rather than JRA-55.

Line 217: 2.5 Mass flux estimates of CFC-11 (this labeled 2.4 in the text)

Figure 4: What are the "two corresponding time series of tropospheric CFC-11 mixing ratios"? The grey solid line represents the NOAA measurements but it's not clear what the grey dashed line represents.

[Figure]

---

## Author Comment (AC1) · 13 Jun 2020

Response to Anonymous Referee #1 (RC1).

General comments

This paper presents novel AirCore measurements of several halocarbons in the stratosphere, and analyses trends in stratospheric trace gases, particularly focusing on CFC-11. The paper is interesting, timely, and the measurement technique shows great promise. I think the paper should be published in ACP, following some (mostly minor) changes.

[Figure]

Author response

We thank the referee for taking the time to review this manuscript in detail and particularly appreciate the many constructive suggestions.

Reviewer comment

In general, I think there is quite a lot of material in the supplement that should be in the main text. Particularly given that one of the main purposes of this paper is to introduce a new measurement technique, I would encourage the authors to move the experimental details to the main paper, and some of the validation. I think that Tables S2 and S3 should be in the main text, as S2 demonstrates the repeatability of this new technique (which is referred to as a key conclusion of the paper), and the results of S3 are discussed at some length in the main text. Similarly, the brief description of the model setup could also be in the main text, as it is important for readers not familiar with this particular model.

Author response

We agree and have moved Tables S2 and S3 to the main manuscript. The first paragraph on CLaMS in the Supplement was moved to the end of the Methods section in the main manuscript:

"The Chemical Lagrangian Model of the Stratosphere (CLaMS) is a Lagrangian chemical transport model, with advective transport calculated from three-dimensional forward trajectories and an additional parameterisation for small-scale turbulent mixing (McKenna et al., 2002). Potential temperature is used as vertical coordinate throughout the stratosphere with vertical velocity estimated from the total diabatic heating rate. Further model details and the chemistry scheme used are described in Pommrich et al. (2014). For the simulations used in this study CLaMS was driven with horizontal winds and diabatic heating rates from three alternative meteorological reanalysis data sets: ERA-Interim (from European Centre for Medium-Range Weather Forecasts, ECMWF),

JRA-55 (from Japan Meteorological Agency), and MERRA-2 (from NASA)."

The relevant section on AirCores was expanded by moving over some information from the Supplement. Previous version:

"The balloon-based AirCore technique was developed further mainly through the use of specially-designed tubing that maximises the amounts of air collected in the stratosphere, as well as through a novel subsampling technique that minimises the use of contamination-prone materials. The amount of retrievable stratospheric air is however still more than two orders of magnitude smaller than from aircraft-based sampling techniques. With laboratory analytical improvements compensating for this, the AirCore measurements show good precisions (ranging from 0.2 to 3.3 % compared with 0.4 to 1.1 % for aircraft samples) and excellent agreement with the aircraft data. More details can be found in the supplement. All samples were processed with a previously described analytical system and methodology (Laube et al., 2010 and 2012,) using cryogenic extraction and pre-concentration, followed by gas chromatographic separation and detection with a high-sensitivity mass spectrometer."

Modified version:

"The balloon-based AirCore technique was developed further mainly through the use of specially-designed tubing that maximises the amounts of air collected in the stratosphere, as well as through a novel subsampling technique that minimises the use of contamination-prone materials. The amount of retrievable stratospheric air is however still more than two orders of magnitude smaller than from aircraft-based sampling techniques. With laboratory analytical improvements compensating for this, the AirCore measurements show good precisions (ranging from 0.2 to 3.3 % compared with 0.4 to 1.1 % for aircraft samples, see Table 1) and excellent agreement with the aircraft data. The other important challenge for AirCore measurements of halocarbons is to ensure that the air is not contaminated throughout the entire sampling and sub-sampling process. Contaminations can arise from leakages and/or halocarbon-emitting materials

(such as organic polymers) in the AirCore itself, in the CO2-analyser system including the pump, or in the subsampling system. Importantly, for all compounds reported here mixing ratios in the stratosphere are much lower than in even remote tropospheric regions, let alone near sources of these gases. In addition, almost all of the contamination possibilities would affect the entire profile as an AirCore is essentially one air sample. This would become apparent in the correlations of the species with each other, which are very compact in the stratosphere. In the absence of such correlation breakdowns (See Figures 1, 2, and S1 to S4) we therefore conclude that such contaminations are at undetectable levels in the data set presented here. More details can be found in Table 1 and the supplement."

Reviewer comment

The title needs to mention that changes in halocarbons are being referred to.

Author response

We have changed the title to "Investigating stratospheric changes between 2009 and 2018 with trace gas data from aircraft, AirCores, and a global model focusing on CFC-11".

Reviewer comment

L34 – 35: "required" is used twice in the same sentence

Author response

Thank you! The first one has been removed.

Reviewer comment

L40: this first line is too vague. It doesn't really say anything. I suggest starting with the current second line.

Author response

We disagree that one of the main motivations for this work (i.e. the fact that many trace gases are strong greenhouse gases and/or enhance the depletion of stratospheric ozone) "doesn't really say anything". We have however changed the beginning of the sentence to specify that we are referring to "halogenated trace gases" here.

Reviewer comment

L42: "low to very low" could just be replaced by "low"

Author response

Done.

Reviewer comment

L45: "upper limit is around 50km"? Is this suggesting that the stratosphere extends to 50km? Not clear as written.

Author response

The statement was changed to "This is modest considering that the stratosphere extends to around 50 km."

Reviewer comment

Section 2 heading: "Methods" are a sub-section of "Results"? I suggest separating these two sections.

Author response

We agree with the referee and have created a new section "2 Methods". The new section 3 has been renamed to "Results and discussion" to better reflect its content and now starts with sub-section "3.1 Observational data overview and comparisons".

Reviewer comment

L75 and elsewhere: I'm not 100% sure about the grammar here, but I would have

thought that a colon followed by a list separated by commas would be more appropriate here (after "platforms")

Author response

We agree and have replaced the first comma with a colon and also removed the "i.e."

Reviewer comment

L80 – 83: As mentioned above, I think this needs much more explanation, and I would bring in a lot of the supplementary information.

Author response

As outlined in our response to comment 2 by RC1 we have followed this advice.

Reviewer comment

L84: "good precision" needs quantifying. Perhaps compare to the precision of other measurement techniques and say whether it is comparable or not (avoid subjective terms like "good").

Author response

We agree with the referee and have added the following after "good precisions" to be more quantitative: "(ranging from 0.2 to 3.3 % compared with 0.4 to 1.1 % for aircraft samples)".

Reviewer comment

L88: AoA is defined later on (L125). Needs to be introduced here. Furthermore, I would suggest a brief paragraph here explaining the concept, and calculation method.

Author response

We have replaced "AoAs" with "mean Ages of Air (AoAs, i.e. average stratospheric transit times, see section 3.1 for more details)".

Reviewer comment

L90: I found this description too hard to follow. When were second or third order polynomials used? What exactly was being fitted? What is the purpose of "quintupling" the data, etc.? I think you need to walk the reader through this more gently. A figure may help to demonstrate the technique.

Author response

We agree that some crucial information was missing here. The original statement

"Stratospheric trends at AoA surfaces were derived by using second- and third-order polynomial fit functions to interpolate onto these surfaces for each flight. To test the uncertainty of this method, the data for each flight was first quintupled by adding and subtracting the mixing ratio and mean age uncertainties and then drawing 500 random samples (repeats possible) at each AoA surface with a bootstrap algorithm (as in Laube et al., 2013). The derived mixing ratios were subsequently used to produce linear regressions over time, including a weighting by the inverse uncertainties of the individual CFC mixing ratios. The bootstrapping algorithm was used again to ensure that the derived slope uncertainties were not underestimated and that individual high or low points did not bias the slope estimates."

was modified and expanded to:

"Stratospheric trends at AoA surfaces were derived by fitting second- and third-order polynomials (depending on whether an inflexion point was observed) to the respective correlations of mixing ratios and AoAs. The formulas of the polynomials were then used to interpolate onto the AoA surfaces (1, 2, 3, and/or 4 years, depending on which AoA range was covered) for each flight. To test the uncertainty of this method, the data for each flight was first quintupled by adding and subtracting the mixing ratio and mean age uncertainties. This resulted, for each data point, in the average plus minimum and maximum value for both mixing ratio and AoA. Subsequently $5n$ ($n$ being the number

of data points available for each flight) random samples were drawn (repeat draws possible) with a bootstrap algorithm as in Laube et al., 2013, and a second or third order polynomial again fitted. This procedure was repeated 500 times for each flight resulting in an average mixing ratio and an uncertainty range at each AoA surface. The derived mixing ratios were subsequently used to produce linear regressions over time, including a weighting by the inverse uncertainties of the individual CFC mixing ratios. The bootstrapping algorithm (500 repeat draws, repeat draws possible) was used again to ensure that the derived slope uncertainties were not underestimated and that individual high or low points did not bias the slope estimates."

With the added explanation we do not feel that an additional figure is required to explain this approach.

Reviewer comment

L134: what does "times inside the stratosphere" mean? Do you mean AoA?

Author response

Yes. "(AoAs)" was added after "times" to clarify this.

Reviewer comment

L174: Given that this line is at the start of a paragraph, I think you need to re-state what each part of the sentence refers to. I.e. a positive trend in what? Explain "all observation based cases".

Author response

We have altered the sentence from

"While there is substantial variability we do find a positive trend (increases from 3 to 10 %) from 2009 to 2018 in all observation-based cases."

to

"While there is substantial variability of mixing ratios at these AoA surfaces over time, we do find a positive trend (increases from 3 to 10 %) from 2009 to 2018 for all observation-based (aircraft and AirCore) estimates."

Reviewer comment

Figure 3: The conclusions drawn from this analysis, namely that there has been no trend, or a positive trend, seems highly dependent on a small number of points in 2009 – 2011. Can the authors comment on this? Are we really confident that the observations can provide strong conclusions about the overall trend, given that there are large periods of these time series with no data (i.e. nothing between 2011 and 2016)? Some softening of the discussion of the observation-based trend would seem warranted to me.

Author response

We agree and have added the following statement to the end of the relevant paragraph:

"However, it should be noted that the limited temporal and spatial coverage of the observation-based and especially the gap between 2011 and 2016 introduces an additional uncertainty that is only partly reflected in the derived slope uncertainties."

In addition, we would like to point out that the following two statements were already included in the Conclusions:

"In a qualitative sense, and keeping in mind the regional nature of these measurements and the uncertainties related to the calculation of stratospheric transport times, this would point towards increasing mass fluxes of CFC-11 being transported back to the troposphere."

"However, any further quantification of the stratospheric part of the CFC-11 story is prevented firstly by the non-global and intermittent nature of sufficiently precise observations. . ."

Reviewer comment

L175: as stated above, having Table S3 in the main text seems important to understand this sentence.

Author response

As outlined in our response to comment 2 by RC1 we have followed this advice.

Reviewer comment

L187: For clarity, I suggest adding: "In contrast, in the stratosphere, we find: : :"

Author response

Done.

Reviewer comment

L195 – 204: I must admit to finding it very challenging to follow this explanation, despite reading over it three or four times. I realise it's a complicated business, but I suggest that the authors re-word, or, better, provide a schematic outlining their argument.

Author response

We feel that a schematic or figure would give too much weight to this small and speculative part of the manuscript. We have however rephrased the explanation to explain the possibility better:

Original paragraph: "The only straight-forward possibility to generate positive CFC-11 trends in the stratosphere between 2009 and 2018 would be an increase in the air fractions that have younger and older residence times than the inferred mean age. If the older air has already lost virtually all of the shorter-lived gases (H-1211 and CFC-11), but not the longer-lived ones (CFC-12 and HCFC-22), then an increase of its share should lead to a decrease of CFC-12 (but not necessarily HCFC-22 which is much longer-lived) without being accompanied by a decrease in H-1211 or CFC-11.

To maintain a similar mean age, the younger fraction of the air would also need to have an increased share, leading to generally higher mixing ratios of all gases – and disproportionally so for HCFC-22 as it continues to increase. Such a change to the stratospheric transit time distributions could be considered as the simplest case that would qualitatively explain our observations."

Changed to: "The only straight-forward possibility to generate positive CFC-11 trends in the stratosphere between 2009 and 2018 would be an increase in the air fractions that have younger and older residence times than the inferred mean age. Such a two-fold increase would maintain the same AoA, but would influence the mixing ratios observed at the AoA surfaces in different ways. If the increased older air fraction had been in the stratosphere for long enough, it would have already lost virtually all of its content of shorter-lived gases (H-1211 and CFC-11). However, if this older air fraction at the same time would be in an AoA range where the longer-lived gases (CFC-12 and HCFC-22) are still present in significant amounts, then an increase in its share should lead to a decrease in CFC-12 and HCFC-22 mixing ratios (but less so for the latter as it is much longer-lived in the stratosphere). To balance this increase in the older air fraction and maintain a constant mean age, the younger fraction of the AoA spectrum would also need to have an increased share. Younger air generally contains higher mixing ratios of all four gases – and disproportionally so for HCFC-22 as its tropospheric mixing ratios continue to increase. If the increases in the two fractions of the AoA spectrum would be in the right AoA range, the overall effect would then be an increase of mixing ratios of CFC-11, H-1211, and HCFC-22 over time at a given AoA surface, accompanied by a decrease in CFC-12 mixing ratios. This would then be entirely consistent with the changes we observed at almost all AoA levels between 2009 and 2018. Therefore such a change to the stratospheric transit time distributions could be considered as the simplest case that would qualitatively explain our observations."

Reviewer comment

L197: I suggest: ": : : HCFC-22, which is much longer lived in the stratosphere" (it's

shorter lived overall).

Author response

Agreed and done.

Reviewer comment

Section 2.4: I'm sure this highlights my ignorance, but doesn't it seem counterintuitive to have an increased mass flux from the stratosphere to the troposphere at the same time as an increase in stratospheric CFC-11? Can you add a couple of sentences explaining why this would be?

Author response

An increased mass flux from the stratosphere to the troposphere would likely be accompanied by an increased troposphere-to-stratosphere mass flux. If the latter increase would not go through the main CFC-11 sink region in the tropical lower stratosphere it could happen at the same time as stratospheric mixing ratio increases of CFC-11. One possibility here would be an increased fraction of air reaching the northern hemispheric stratosphere through the Asian Summer Monsoon. The following statement was added to explain this possibility:

"Such a flux increase could be consistent with the observed increases in CFC-11 mixing ratios on AoA surfaces (Section 3.3) if accompanied by an increased fraction of air entering the stratosphere without passing through the main CFC-11 sink region in the lower tropical stratosphere (and instead entering e.g. through the Asian Summer Monsoon)."

Reviewer comment

L230: "ratio", rather than "ratios"

Author response

Changed, thanks!

Reviewer comment

Figure 4 caption: Specify which direction the flux is in (I think it strat to trop?).

Author response

It is indeed and we have added that.

---

## Author Comment (AC2) · 13 Jun 2020

Response to Anonymous Referee #2 (RC2).

Reviewer comment

This paper is very well written and well referenced. It makes important contributions to; 1) evaluating comparisons of halocarbon measurements from different sampling platforms over a 10 year period, 2) the use of halocarbon measurements to evaluate stratospheric dynamics in a global model using different meteorological reanalyses, and 3) exploring the use of model derived atmospheric dynamics to examine reasons

for the reduction in CFC-11 global mixing ratio decreases.

Overall this paper is high quality and should be accepted after addressing the few suggestions noted below.

Author response

We thank the referee for reviewing this manuscript and very much appreciate the constructive suggestions.

Reviewer comment

Line 41: The phrase "allow to gauge" is awkward and should be revised.

Author response

We have changed this to "allow to constrain".

Reviewer comment

Line 88: This is the first use of AoAs and it should be spelled out.

Author response

"AoA" was changed to "Ages of Air (AoAs, i.e. average stratospheric transit times, see section 3.1 for more details)".

Reviewer comment

Line 90: Polynomial fit functions of what? I suggest adding in "of AoA vs mixing ratio of each species" or something like that.

Line 93: The bootstrap method isn't explained fully in Laube et al., 2013 so I suggest including Volk et al., 1997 with the Laube et al., 2013 reference here. A bit more explanation would also be helpful.

Author response

The explanation and reference have been added, alongside with an expanded explanation of the methodology, please see our response to RC1 (comment on L90).

Reviewer comment

Line 108: I suggest including a reference to Figures S5 and S6 here.

Author response

Good idea! This has been added.

Reviewer comment

Line 112: I suggest referencing Table S2 for the measurement uncertainties.

Author response

Done. As outlined in our response to comment 2 by RC1 we have also moved Table S2 to the main manuscript.

Reviewer comment

Lines 114-115: The authors state here that Figure 1 illustrates improved temporal density from 2016, which it does, but goes on to say especially at altitudes above 15 km. There is no indication of altitude in Figure 1. As the authors mention, the lower values are from higher altitudes, but there is nothing that says what those altitudes are, and in fact, the values in Figure 1 are not from the highest altitudes: : :see next comment re Figure 1.

Figure 1: The mixing ratios for all compounds do not reflect the full range of the combined measurements. The full range for CFC-11 and CFC-12 are seen in Figure 2 and for the other compounds in Figures S1-S4. Please indicate in Figure 1 why this is the case.

Author response

We agree and have modified the caption of Figure 1 to include the relevant information:

"For all gases except SF6 some higher altitude data are not shown to better demon-strate the good comparability of near-tropopause data to the NOAA time series. The complete corresponding data including uncertainties can be found in the supplement (see also Figures S1 to S4)."

Reviewer comment

Line 146: The authors state the MERRA-2 based data stands out producing higher transport times at similar stratospheric CFC-11 mixing ratios. It should read "higher mean ages", rather than higher transport times. As the authors say in line 148, this results from slower transport times.

Author response

Changed, thanks!

Reviewer comment

Line 176: I suggest including Figure S7 in the main text so the readers can see all four years rather than going to the Supplementary section for 2 of the 4 years. Line 1 S12 and S13 Figure captions should be MERRA-2 rather than JRA-55.

Author response

We agree and have implemented the requested changes. Figure S7 is now part of Figure 3.

Reviewer comment

Line 217: 2.5 Mass flux estimates of CFC-11 (this labeled 2.4 in the text)

Author response

This has been adjusted alongside with the restructuring of the section in response to RC1 (comment on the Section 2 heading).

Reviewer comment
* * *
Interactive
comment

Figure 4: What are the "two corresponding time series of tropospheric CFC-11 mixing ratios"? The grey solid line represents the NOAA measurements but it's not clear what the grey dashed line represents.84: Please reference Figures S8-S13 for this discussion.

Author response

The respective statement in the figure caption was modified to "Shown in grey and on the right hand y-axis are the two corresponding time series of tropospheric CFC-11 mixing ratios (i.e. the real one, solid, and the one with the forced decrease, dashed)."

---

## Author Response (AR2)

Referee comment

1. The title. I still think the title needs to mention "halocarbons" as the subject of the "changes" that are being referred to (I think my previous comment may not have been specific enough on this point). Perhaps "Investigating changes in stratospheric CFC-11 and other halocarbons using aircraft, AirCores and a global model"?

Author response

We have changed the title from

"Investigating stratospheric changes between 2009 and 2018 with trace gas data from aircraft, AirCores, and a global model focusing on CFC-11"

to

"Investigating stratospheric changes between 2009 and 2018 with halogenated trace gas data from aircraft, AirCores, and a global model focusing on CFC-11"

Referee comment

2. The new paragraph on AoA surfaces is much improved. However, I still get a bit stuck on the "quintupling"! I think what the authors saying is this: "… the data for each flight were replicated four times, where each replicate was modified by plus or minus the uncertainty in the mixing ratio and mean age uncertainties"?

Author response

We have changed the sentence as requested and thank the reviewer for this suggestion.

Referee comment

3. I think there's a missing word from the new sentence: "However, it should be noted that the limited temporal and spatial coverage of the observation-based and especially the gap between 2011 and 2016 introduces an additional uncertainty that is only partly reflected in the derived slope uncertainties." I'm not sure what this should be! I think this sentence needs rewording.

Author response

We have improved the sentence by changing it from:

"However, it should be noted that the limited temporal and spatial coverage of the observation-based and especially the gap between 2011 and 2016 introduces an additional uncertainty that is only partly reflected in the derived slope uncertainties."

to

"However, it should be noted that the limited temporal and spatial coverage of the observation-based and especially the gap between 2011 and 2016 represents an additional and not quantifiable source of uncertainty."